# Respiratory Rate Estimation from Thermal Video Data Using Spatio-Temporal Deep Learning

**DOI:** 10.3390/s24196386

**Published:** 2024-10-02

**Authors:** Mohsen Mozafari, Andrew J. Law, Rafik A. Goubran, James R. Green

**Affiliations:** 1Department of Systems and Computer Engineering, Carleton University, Ottawa, ON K1S 5B6, Canada; mohsenmozafari@cmail.carleton.ca (M.M.); andrew.law@nrc-cnrc.gc.ca (A.J.L.); goubran@sce.carleton.ca (R.A.G.); 2Flight Research Laboratory, National Research Council of Canada (NRC), Ottawa, ON K1A 0R6, Canada

**Keywords:** respiration rate estimation, thermal video, deep learning, face detection

## Abstract

Thermal videos provide a privacy-preserving yet information-rich data source for remote health monitoring, especially for respiration rate (RR) estimation. This paper introduces an end-to-end deep learning approach to RR measurement using thermal video data. A detection transformer (DeTr) first finds the subject’s facial region of interest in each thermal frame. A respiratory signal is estimated from a dynamically cropped thermal video using 3D convolutional neural networks and bi-directional long short-term memory stages. To account for the expected phase shift between the respiration measured using a respiratory effort belt vs. a facial video, a novel loss function based on negative maximum cross-correlation and absolute frequency peak difference was introduced. Thermal recordings from 22 subjects, with simultaneous gold standard respiratory effort measurements, were studied while sitting or standing, both with and without a face mask. The RR estimation results showed that our proposed method outperformed existing models, achieving an error of only 1.6 breaths per minute across the four conditions. The proposed method sets a new State-of-the-Art for RR estimation accuracy, while still permitting real-time RR estimation.

## 1. Introduction

Physiological signals provide rich information about one’s health condition. Vital signs, such as respiration rate (RR) and heart rate (HR), can be estimated from such signals and used to detect underlying changes in health status. RR measurement is usually conducted using a respiratory effort belt, whereas HR is recorded using electrocardiography electrodes or a pulse oximeter [1]. These sensors require body contact, which can cause discomfort for the subject, reduced mobility, increased risk of infection/contagion, and skin–electrode interface artifacts. Non-contact physiological signal monitoring is, therefore, preferable in long-term monitoring or remote health screening, but this requires advanced analysis based on signal processing and machine learning to extract reliable vital sign estimates [2].

Non-contact vital sign measurements can be achieved through analyses of videos recorded using depth [3], color (RGB), or thermal [4] imaging modalities. Such methods typically use different regions of interest (ROI) to focus the signal estimation model on informative areas in the video frame. These areas are sometimes defined manually. For example, Bennet et al. [5] used a pre-defined ROI of the forehead in RGB recordings for HR estimation, while Ruminski et al. [6] used the nostril area in a thermal video to estimate RR. These areas were chosen to exclude irrelevant information in the video data.

In a fully automated estimation model, the ROI must be detected automatically [7]. These deep learning (DL) computer vision models are usually pre-trained on large RGB image datasets, such as COCO [8], to learn common features in image processing tasks. ROI detection models are then fine-tuned [9] on the target domain (e.g., facial ROI detection) to automatically perform the spatial cropping of a video to focus on relevant information within the frame. The physiological data must be extracted and processed from the cropped video to robustly estimate vital signs information.

The most direct way to derive a one-dimensional (1D) physiological signal from a 3D video (time × width × height) is through the spatial averaging of the pixel values inside the ROI. For example, Chauvin et al. [10] used a pan–tilt thermal camera to track the mouth and nose area, and then estimated the respiratory signal by spatial averaging across the ROI. The mouth and nose areas were chosen because of the respiration effect on these regions. Respiration warms and cools these areas during exhalation and inhalation, respectively, which can be captured by a thermal camera.

For HR estimation from RGB videos, Shan et al. [11] used a forehead ROI because they found that using an ROI from areas below the eyeline led to greater HR estimation noise. Their method was based on an independent component analysis (ICA) to extract physiological sources, and the HR was calculated from the power spectral density of the desired signal. Poh et al. [12] proposed an ICA-based method to estimate the PPG signal from RGB videos. The spatial average of each RGB channel was calculated, then all three signals underwent ICA. Finally, the ICA-extracted source that was most relevant to the PPG signal was chosen to estimate HR. In [13], Chen et al. used an RGB face video to estimate respiratory signal. The signal, which was extracted through averaging the pixels, was processed by motion compensation and filtering to reduce the noise and obtain a more reliable respiratory signal. In another study, Alnaggar et al. [14] extracted the respiratory signal from chest movements. Before extracting the signal, an RGB video was processed using Eulerian video magnification to keep the desired frequency information.

In another study, Kowalczyk et al. [15] investigated respiratory signal estimation using thermal videos of subjects who were wearing masks. In their study, a YOLO-based model was used to detect the mask ROI in each frame, and the spatial average of all the pixel values within the mask region in each frame created an estimated respiratory signal over time. Choi et al. created a model that uses a multi-resolution window, phase sensitive processing, and a bi-directional LSTM to find the respiratory rate [16]. However, this model was evaluated on patients undergoing surgery, with no movement, and used a fixed thermal camera that was focused on the ROI. In a condition of having no subject movement, the respiration rate estimation problem is simplified. Cho et al. used a respiration tracking algorithm to extract the respiratory pattern from the nostril area. The final goal of their study was to detect the stress levels of the subjects, which was calculated according to their breathing patterns, using a convolutional neural networks (CNN) model [17]. Lyra et al. [18] used a CNN to automatically detect the chest ROI in the thermal videos and to extract the respiration signals of ICU patients. Kwasniewska et al. [19] used a CNN model to enhance low resolution thermal videos of the nose region prior to estimating respiration rate.

In addition to ROI detection and spatial averaging methods, some studies have proposed end-to-end DL-based approaches that directly estimate a physiological signal from video input. Chen et al. proposed DeepPhys [20], which is an attention-based approach, to recover HR and RR from RGB video data. CNNs process two streams of data: (1) the current frame and (2) the difference between successive frames. The spatial attention mask is created by examining the current frame (Stream 1) and indicates which pixels have greater importance in the frame difference stream (Stream 2). Spatial information within each frame is processed using 2D CNNs; however, this approach does not directly model the dependencies that may exist between the spatial information and information in the time axis.

In response to this limitation, Yu et al. introduced PhysNet [21], which leverages 3D CNNs to simultaneously process temporal and spatial changes in the video. This model achieved promising results for HR estimation using RGB videos. The estimated signal is output directly from the 3D CNN blocks using a negative Pearson correlation loss function. However, 3D CNN models have yet to be examined for RR estimation, particularly from thermal videos.

In addition, 3D CNNs were also used in iBVPNet [22] to estimate HR from RGB videos. The iBVPNet architecture can be seen as a temporal encoder–decoder model, which processes the spatial data simultaneously. iBVPNet was trained using the cosine similarity (CS) loss function because they found that it outperformed negative Pearson correlation.

Kumar et al. introduced StressNet [23], which analyzes thermal videos to estimate initial systolic time intervals, providing an indication of the stress level from cardiac activity. Rather than using 3D CNN for integrated spatio-temporal processing, StressNet performs spatial processing using a 2D ResNet-50 CNN model [24], which is pre-trained on ImageNet. Temporal processing is achieved through the sequential processing of spatially processed videos using long short-term memory (LSTM) DL blocks.

In some studies, multiple video signals have been used to estimate a physiological signal. For example, Jaiswal et al. [25] introduced FuseNet, which fuses RGB and Multi Scale Retinex (MSR) images to extract HR from facial videos. FuseNet’s architecture consists of three parts: (1) preprocessing, (2) feature mapping, and (3) CNN blocks for estimating HR.

While these recent advances demonstrate the promise of end-to-end DL approaches in vital sign estimation from videos, these methods generally focus on RGB videos (for which much pretraining data are available) and HR estimation, rather than RR estimation from thermal videos. The use of thermal video data are preferred to RGB due to its privacy preserving qualities [26] and the fact that it can detect breathing-related temperature changes in addition to movement information. Therefore, we were highly motivated to create an end-to-end deep learning approach for respiratory signal estimation from thermal video data. Our proposed approach is motivated by the successful application of ROI detection in [11,12]; the use of 3D-CNN for integrated spatio-temporal processing, as used in [21]; and the use of LSTM for temporal post-processing, as used in [23].

In this paper, which builds on our recent work on thermal face detection using DL [27], we propose an end-to-end DL model for respiration signal estimation from thermal videos. Considering that there is interdependency between spatial and temporal information within a video, using 3D CNNs enables the simultaneous processing of spatio-temporal aspects. However, shallow 3D CNNs cannot fully capture long-term temporal dependencies, while deeper 3D CNN models result in higher computational complexity. Our new model solves this problem by using a shallow 3D CNN stage followed by an LSTM stage, which can strengthen the long-term temporal processing. In particular, bi-directional LSTM models are used to examine both forward and reverse temporal relationships within a window of thermal video data.

Another challenge in training video-based models using respiratory effort belt data is the inherent asynchrony. A thermal face video captures changes in temperature around the nose and mouth of the subject. However, a respiratory belt captures changes in chest or abdomen circumference, which are offset in time, relative to peak temperature fluctuations at the nose and mouth. Existing loss functions for regression learning do not account for this time lag issue. Therefore, we have developed a novel loss function, which considers the maximum cross-correlation between contact sensors and video data, accounting for the temporal lag between the two signals. Additionally, the difference in peak frequency within the respiration frequency band is used as a secondary term in the loss function to ensure that the video-based estimated respiratory signal agrees with the gold standard respiratory signal from the respiratory effort belt. These two novel approaches were implemented to solve the time asynchrony between the reference and predicted respiratory signals.

Figure 1 shows the flowchart of our proposed method. The summary of our proposed method is as follows:

The facial ROI is identified using a Detection Transformer (DeTr) model [28]. To account for the domain shift between RGB pretraining data and thermal video data, transfer learning over a large thermal facial image dataset is conducted prior to fine-tuning the DeTr model using a subset of our own dataset.

The resulting spatially cropped video is processed by a 3D CNN, followed by a bi-directional LSTM stage to produce the final estimate of the respiratory signal.

The estimated RR is calculated by finding the maximum frequency peak within the output signal. The novel loss function described above enables the model to be trained directly using thermal video data with gold standard respiratory signals.

A performance evaluation across 22 subjects indicated that our proposed model out-performed the State-of-the-Art and provided a more robust respiration signal estimation across multiple recording conditions. The PyTorch code for our proposed method can be found on GitHub (https://github.com/MohsenMozaffary/Deep_Respiration).

According to the literature review, the research to date has primarily focused on estimating HR from RGB video data, whereas, here, we estimate the respiration signal from thermal videos. The lower RR, when compared to HR, requires careful consideration of the temporal field depth and longer-term dependencies in the data, for which 3D CNN that was developed for HR estimation may not be appropriate. Therefore, in the proposed approach, an LSTM layer was introduced following the 3D CNN layers to capture longer-term dependencies in the data related to respiration. Moreover, for RR estimation, we expect a phase shift between the recorded video and the ground truth signal from contact sensors. To solve this problem, a new loss function was introduced to account for phase shift.

Ultimately, this paper makes the following research contributions:(1)A fully deep-learning pipeline has been developed for estimating respiration rates from thermal video data. Existing pipelines are largely focused on RGB video and HR estimation.(2)Several video-based deep learning vital sign estimation methods from the literature have been adapted for RR estimation using thermal video data.(3)A thorough performance evaluation of the proposed method indicated an average error rate of 1.6 ± 0.4 breaths per minute. This has exceeded the performance of four State-of-the-Art, video-based vital sign estimation methods, here modified for RR estimation from thermal video data.(4)By using extremely fast facial ROI detection and video cropping, a shallow 3D CNN architecture, and an LSTM stage, the proposed pipeline has a runtime of only 5.1 ms, permitting near real-time updated RR estimates.

## 2. Methods

As shown in Figure 1, the proposed respiratory signal estimation pipeline comprises an ROI detection stage, a 3D CNN stage for spatio-temporal processing within each frame, followed by an LSTM sequential stage for estimating the respiratory signal. This section describes the data used for developing and evaluating the method, followed by each stage of the signal estimation pipeline.

### 2.1. Experimental Setup and Dataset

Following informed consent, data were recorded from 22 participants (18 male/4 female, aged 44.0 ± 12.8 years) using a study protocol that was approved by the National Research Council of Canada (NRC) Research Ethics Board. Thermal videos of the participants were captured using a FLIR^®^ T650sc camera (FLIR Systems, Inc., Wilsonville, OR, USA) with a spectral range: 7.5–14 µm, 30 frames per second, 640 × 480 resolution, and uncompressed format. The participants were positioned 2 m in front of the camera and wore a SleepSense^®^ respiratory effort sensor belt (Piezo Crystal Effort Sensor 1387-kit, S.L.P. Inc., Elgin, IL, USA) to detect changes in their abdominal circumference resulting from respiration. Respiration effort data were recorded at 2048 Hz using a BioSemi data acquisition system (BioSemi B.V., Amsterdam, The Netherlands).

Thermal video and respiratory effort data were captured from each participant under four conditions: (1) sitting with a mask, (2) sitting without a mask, (3) standing with a mask, and (4) standing without a mask. Figure 2 shows examples of thermal video frames in the aforementioned four conditions. For conditions 1 and 2, the seat was mounted on a moveable raised platform, so that the seated position of the head in the thermal video frame was approximately the same as in the standing conditions (3 and 4). For conditions 1 and 3, participants wore a disposable, non-medical mask (InterMask, Quanzhou Haina Hygienic Products Co., Ltd., Quanzhou, China). For each condition, the participants were asked to minimize voluntary movements while facing the camera for 90 s data collection segments. A sliding window (length 10 s, with 5 s overlap) was used to extract video clips from each 90 s recording. It resulted in 1496 video clips, with lengths of 10 s. These clips were used to develop and evaluate the proposed method, avoiding clips from the same research participant in both the training and testing subsets.

This dataset was previously used in [29] to estimate the RR from the whole video frame without DL-based models.

### 2.2. ROI Detection

Most facial ROI detection methods process individual video frames using CNN-based algorithms. Both the accuracy and computational complexity of ROI detection should be considered when forming part of an RR estimation pipeline. In [27], several cutting-edge object detection methods were compared in terms of performance and computational complexity for face detection in thermal videos. It was concluded that detection transformer (DeTr) models [28] outperform existing object detection algorithms for this task. DeTr uses a CNN backbone, positional embedding [30], a transformer encoder [31], and a fully connected layer. The CNN backbone is normally a pre-trained model, such as ResNet50 [32], which is fine-tuned during the task-specific training phase. The processed images produced by the CNN backbone are decomposed into several smaller image patches. The position of each patch within the grid is encoded using positional embedding. A transformer encoder and decoder process the embedded image patches. Finally, a fully connected layer determines the class and bounding box coordinates.

The face detection algorithm examines each frame to find the facial ROI, following a track-by-detection approach. The resulting ROI-cropped images are stacked to create a dynamically cropped video. Any frame-to-frame variability in the detected ROI coordinates causes a jitter in the resulting stacked video, which reduces the accuracy of downstream respiration estimation. To solve this problem, the DeTr network was first fine-tuned on the large TFW dataset [33], which contains thermal images of human faces. The resulting model was again fine-tuned using our dataset, which has 2700 samples from 18 subjects. By leveraging two fine-tuning datasets, the facial ROI frame-to-frame stability was substantially improved [27]. Moreover, the fine-tuned DeTr-based facial ROI detection model processed 32 frames in 39 ms [27], enabling a tracking-by-detection approach, where the facial ROI is detected continuously in each frame, maintaining a sampling rate of >1 Hz.

### 2.3. DL Respiratory Signal Estimator

The next stage examines the sequence of the 2D cropped thermal images produced by the ROI detection stage and estimates the 1D respiratory signal. To achieve this goal, 3D CNN (spatio-temporal) models are used to process the 3D video data (time × width × height). The time dimension of the video data is expected to reflect respiratory patterns. To capture such time dependencies, Lea et al. introduced temporal encoding and decoding, using temporal pooling and upsampling, respectively [34]. By considering [21,34], we adopted a temporal encoder–decoder 3D CNN model architecture. In a vanilla 3D CNN block, average pooling is applied only to the two spatial dimensions and has no effect on the sequence length (time dimension). However, in an encoder 3D CNN block, average pooling is applied also to the time dimension, which reduces not only the spatial size, but also the length of the input sequence/volume [21]. Likewise, in the decoder block, the transpose 3D CNN uses a stride greater than one, thereby expanding and restoring the original sequence length. Additional vanilla 3D CNN blocks are used to further process and reduce the dimension of frames. Figure 3 illustrates the encoder, decoder, and vanilla 3D CNN blocks used in the present study.

Although in [34] it was mentioned that 3D encoder–decoder blocks can capture time dependencies, very long dependencies cannot easily be captured through 3D CNNs unless many layers are used, which would increase computational complexity. To solve the problem of long-term sequence processing, we used a many-to-many bi-directional LSTM layer. A bi-directional LSTM layer was used because we could process a 10 s sliding window of thermal video data in both the forward and reverse directions to better estimate the respiratory signal. In the final stage of the signal estimation model, a fully connected layer connects the results from the LSTM stage to the output respiratory signal.

To focus on respiratory signal activities, the estimated signal from the model was post-processed using a third order IIR filter in a frequency band of 0.05 Hz to 0.7 Hz (corresponding to respiration rates between 3 and 42 breaths per minute).

The final RR estimate is calculated by finding the frequency with peak power in the frequency spectrum of the resulting reconstructed signal. Figure 4 illustrates the complete model architecture.

### 2.4. Loss Function

The recorded video data and ground truth signal represent the respiratory activity measured from two separate areas. Thermal video respiratory data arise from heat changes around the nose and mouth of the subject (or the face mask area). However, the ground truth signal arises from changes in the circumference of the abdomen of the subject. We can expect an asynchrony between the torso volume and facial heat changes [35]. This phase shift can reduce the effectiveness of a loss function, such as the root mean square error (RMSE) between two signals, which is a widely used loss function for regression. To account for the expected asynchrony between predicted and target signals, the negative maximum cross-correlation (NMCC) was used as a core term in the loss function.

In order to efficiently calculate the maximum cross-correlation between signal *A* and *B*, firstly, we need to find the fast Fourier transform (FFT) of the signals. Let us call the FFT of *A* and *B* signals $A^{\prime }$ and $B^{\prime }$, respectively. Also, the conjugate of $B^{\prime }$, denoted as $B^{\prime *}$, is computed by negating the imaginary part of $B^{\prime }$. The convolution of the two signals of A and B in the frequency domain can be calculated by $A^{\prime }$ × $B^{\prime *}$. By taking the inverse FFT of convolved signals we can find the cross-correlation of the original signals of *A* and *B*. This output will show the similarity between the two signals of *A* and *B* with different lags. By also filtering in the frequency domain, the cross-correlation in a specific frequency range can be computed. A respiratory signal typically lies between 3 breaths per minute (BPM) for a meditating condition, and 42 BPM for a high breathing rate, which means the frequency range of interest is between 0.05 Hz and 0.7 Hz. Therefore, we can filter $A^{\prime }$, $B^{\prime }$ to compute the cross-correlation in this frequency range of interest. Equation (1) shows the calculation of cross-correlation for *A* and *B*:(1)$$x-\mathrm{corr}=f\left(A^{\prime }\right)\times {f(B\prime )}^{*}$$
where *f* is a bandpass filter in the range of 0.05 to 0.7 Hz. The negative of the maximum value of *x-corr* is used in the NMCC loss function.

Additionally, for a correctly estimated respiratory signal, the spectral peak of the target and predicted respiration signals should be similar so that the correct RR can be estimated from the frequency with peak power. To include this concept in the loss function, we calculated the absolute frequency difference (AFD) between the peak power frequencies of the ground truth and predicted spectra within the frequency range of interest (0.05 Hz to 0.7 Hz). An AFD value of more than 3 BPM was undesirable, therefore, the observed absolute difference was mapped to the (0, 1) range, saturating at 3 BPM. A predicted signal completely aligned with ground truth will have an AFD of 0 and any difference ≥ 3 BPM is defined as an AFD of 1.

Considering that the respiratory signal within the video data can have phase shift compared to the ground truth signal, and that there may be noise in either the video or the ground truth respiration signal, a combination of NMCC and AFD could help the model to overcome problems related to such sub-optimal recordings. The final loss function is the combination of NMCC and AFD, as defined in Equation (2):(2)$$Loss=\alpha \times NMCC+(1-\alpha )\times min(1,\frac{AFD}{3})$$
where *α* is a constant related to the importance of NMCC function. In a situation where noise is expected to be dominant, we can consider using a smaller *α*, which will result in greater focus on the AFD term. In a condition where we are confident about the ground truth signal and video data quality, we can place greater focus on NMCC. In our model, we decided to place equal importance on both NMCC and AFD. Therefore, we used *α* = 0.5. However, in a condition in which data quality is not characterized, we can use hyperparameter optimization approaches to tune α, such as Bayesian optimization [36] or random search [37].

### 2.5. Implementation Considerations

The proposed model was trained 5 times on different sets of subjects (each set included 18 training, 2 validation, and 2 test subjects), and the average performance reported on the results. Each subject had four videos corresponding to the conditions of sitting masked, standing masked, sitting not mask, and standing not masked. In all cases, different research participants were used to train and test the models.

The video sampling frequency was 30 fps, which can correctly capture frequency information up to 15 Hz, according to the Nyquist criterion. Considering that the respiratory signal was less than 1 Hz, the thermal video frame rate was safely down-sampled to 5 Hz to reduce computational costs. Each video sample had a length of 10 s, which led to 50 frames. Also, each frame was resized after face detection to have dimensions of 128 width and 128 height.

The signal estimation model was trained for 15 epochs, with a batch size of two. The Adam optimizers were used with a learning rate of 0.001. The models were trained on a Core i7 PC with 32 GB RAM and a 24 GB 3090 NVIDIA GPU (NVIDIA, Santa Clara, CA, USA).

### 2.6. Comparison with the State-of the-Art

To compare our proposed method of RR estimation from thermal video data with the State-of-the-Art in video-based vital sign estimation, four methods were adapted and evaluated as follows: (1) The face tracking and spatial averaging method of Kowalczyk et al. [15] was reimplemented and evaluated as a non-DL baseline approach. (2) DeepPhys, which was originally developed for HR and RR estimation from RGB video, was modified slightly to be compatible with thermal videos. The input convolutional layer was reduced from 3-channel (RGB) to 1-channel (thermal video). (3) PhysNet was originally developed for HR estimation from RGB videos. Similarly to DeepPhys, the input 3D convolution layer was first changed to use a single (thermal) channel. Additionally, the time strides for 3D average pooling were modified to account for the different expected frequency ranges of respiration vs. cardiac activity. As mentioned above, the sampling frequency of thermal video data was reduced to 5 Hz from 30 Hz. Given that respiration was expected to be as high as 0.7 Hz, according to the Nyquist theorem, we could not use more than one average pooling layer with a stride of two. Therefore, the PhysNet model was changed to reduce the average pooling and the number of layers in the encoding stage. (4) StressNet was originally developed for cardiac systolic time interval estimation from thermal video data. The input stage of StressNet is a pre-trained ResNet50 backbone model, trained on RGB images. In order to use the learned weights of ResNet50, the first three convolutional layers of the pre-trained ResNet50 backbone model were fine-tuned to extract the respiration-related features. Additionally, ResNet50 needs inputs to have three channels, therefore, each frame was repeated three times to be compatible with the first layers of the model, as in the original StressNet published model [23]. The final State-of-the-Art model included in our comparison was iBVPNet [22]. This model was originally designed to extract HR from either RGB or thermal video data. Considering that this model was designed to work with both thermal and RGB video data, no changes were required to the model architecture to make it compatible with our problem.

All the models were assessed using the following criteria:

Error
in BPM: The ground-truth and estimated signal peak frequencies should agree in
a correct respiratory signal estimation. Therefore, errors in RR can be
calculated by finding the difference between ground truth and estimated
signals’ peak frequency, converted to BPM. The lower error in RR will show a
more reliable model.Max
*x-corr*: Maximum cross-correlation can be a reliable criterion to explore the
model’s reliability in constructing the respiratory signal, while allowing for
the expected phase shift between facial and abdominal breathing artifacts. A
maximum cross-correlation closer to one is more desirable and shows the model’s
capability to estimate the respiratory signal.The
percentage of estimates with less than 2 BPM error: An error that is less than
a specific number can show the performance and reliability of a model [13]. Therefore, we report the percentage of video
clips that result in an absolute error of less than 2 BPM.Inference
time: A respiratory signal estimation model should be sufficiently fast to be
used in real-time scenarios. Therefore, a low inference time is desirable.Learnable
parameters: The number of trainable parameters is also another criterion that
can limit the model’s implementation in real-life scenarios. A model with fewer
parameters is more desirable since it requires less memory and is easier to
deploy on edge devices.

The code for implementing all methods in this paper is available at https://github.com/MohsenMozaffary/Deep_Respiration.

## 3. Results and Discussion

The results for the face detection stage are reported in [27]. Figure 2 shows four examples of annotated faces, with and without masks.

The resulting dynamically cropped video was processed using the signal estimation methods. The performance of the model was measured by comparing the peak frequencies of the predicted and ground-truth signals and converting those frequencies to RR in BPM. Table 1 shows the RR estimation results compared to three State-of-the-Art methods.

We can also see that the maximum cross-correlation was roughly proportional to the error in BPM, which confirmed the relation between the proposed loss function and the final desired outcome, which is the error in BPM.

A typical choice for a loss function in a regression problem is RMSE. However, given that we expect a phase shift between the abdominal gold standard signal and the recorded video, RMSE is not a suitable choice for loss function. Table 2 compares the results for RMSE and our introduced loss function in the proposed model. Clearly, the RMSE loss function did not help the model to converge because it incorrectly assumed that the video-based respiratory and ground truth signal are phase-aligned. The proposed loss function combines NMCC and AFD, which both account for phase shift.

In addition to considering the dataset as a whole, we can also analyze the model’s results in each condition: standing, sitting, wearing a mask, and not wearing a mask. Table 3 reports the results separately for each condition. As expected, based on the expected differences in subject motion (e.g., body sway during standing), the best results were observed for the seated subjects. Although a face detection algorithm can largely account for subject motion, the tracked face will have some uncertainty, which can result in noise in the cropped video, especially for standing subjects.

In terms of mask usage, it is interesting to note that the model performed equally well for subjects wearing and not wearing a mask in both conditions of sitting and standing. This result indicates that our model, including both face tracking and respiratory signal estimation, is robust for face mask usage. The method of Kowalczyk et al. [15] resulted in respiration error in our dataset that was substantially higher than that of the deep-learning-based models. Since body movement activity frequency can be close to respiratory frequency, by performing a spatial averaging, both noise and respiratory activity may be retained in the resulting signal. This may explain the low performance of spatial averaging methods. Signal estimation using spatial averaging across the facial ROI for masked individuals can be used as an RR estimation method. However, their method resulted in an error of 3.5.

Due to the expected phase shift between the gold standard respiratory signal and the video-based estimated signal, RR estimation error is the best indicator of model performance. However, it can be instructive to examine the respiratory signal itself. Figure 5 shows the predicted signals for all models. Only 3D CNN + bi-LSTM; 3D CNN + LSTM; iBVPNet; and the PhysNet (Figure 5a–d) resulted in signals with the correct number of peaks compared to the ground truth signal. Although these peaks are not aligned with the ground truth signal, these signals are sufficiently accurate to correctly estimate the RR. Our proposed approach was motivated by the successful application of ROI detection in [11,12]; the use of 3D-CNN for integrated spatio-temporal processing, as used in [21]; and the use of LSTM for temporal post-processing, as used in [23].

Our proposed method showed that a combination of 3D CNNs for short-term spatio-temporal processing and bi-directional LSTM for long-term dependencies will result in a more robust model than existing models that are using 2D CNNs with LSTM or 3D CNNs alone. Also, it can be concluded that our novel loss function can solve the problem of having a delay between video data and ground truth reference data.

### 3.1. Explaining Model Predictions Using Grad-CAM

Other than the performance of the model, which is reported above, we can also investigate the explainability of the model. Finding the regions on which the model is focusing can provide information about the key spatial respiratory sources within the video. We used Gradient-weighted Class Activation Mapping (Grad-CAM) [38] to examine the model activations from the initial CNN layers to the final CNN layer within the spatio-temporal processing block (see Figure 4), which is the last layer that contains the spatial information from the input image. Using these activations, we visualized the spatial video regions that have the greatest contribution to extracting the final signal. The gradients were averaged and used to create the heatmap of important areas, which was superimposed on a thermal video frame for visualization. Figure 6 shows the focused area for a subject wearing a mask (left), and another subject not wearing a mask (right).

According to Figure 6, in the condition where the subject was wearing a mask, our model focused on the mask area. This can be related to the fact that a mask can absorb the heat from exhaled air and represent the respiratory signal. Other than the mask area, we can also see some secondary areas close to the shoulder, related to respiratory-based movements. In the condition of wearing no mask, the model focused on the nose, the top of the head, and the neck. The nose area could be related to changes in the nose’s skin temperature due to inhalation and exhalation. The neck and top of the head area could be related to the respiratory-related movements.

It can be concluded that our model uses both respiratory-related temperature changes and body movements to extract respiratory signals. Therefore, using thermal video data is expected to result in improved respiratory signal estimation compared to RGB video data.

### 3.2. Performance on Individual Subjects and Conditions

Other than the total performance of the proposed model, we can also investigate the limitations of our model by examining its performance on specific subjects. Investigating the subjects one-by-one, it was concluded that subjects with beards yielded a better performance compared to the subjects without facial hair. This indicates that facial hair provides a surface that absorbs exhaled heat, with such changes apparent in the thermal video.

Comparing different recording conditions, the subjects in a standing condition produced a lower performance compared to sitting subjects. This difference resulted from the body sway, which can mislead the model during spatial processing.

Additionally, wearing a face mask could improve the model’s performance. Having a face mask adds to the surface of heat absorption, which can result in easier respiratory signal extraction for our model. Table 3 shows the results of RR estimation in different conditions.

### 3.3. Limitations

The limitations of our study primarily relate to the high quality of the dataset used for development and evaluation. All data were collected using a high-quality FLIR^®^ T650sc camera, in a controlled environment, with a plain background, and which was free from thermal interference. Research participants were instructed to face the camera and to minimize body movements. The participants were also positioned in such a way that their faces were roughly centered within the video frames.

Considering that the data recording was in a controlled condition, with a small amount of movement, we observed that body sway results in a decreased performance from our model, meaning that we expect to see lower performance in conditions with uncontrolled movements. Therefore, our algorithm will be most performant for remote respiratory signal estimation during relatively controlled conditions, such as patient monitoring in hospital, or passenger monitoring in a vehicle.

Validating the proposed methods on an external dataset would serve to reinforce the observed performance gains. Unfortunately, we were unable to identify a suitable dataset, which included both a thermal video of the face and also the ground truth respiratory signal. There are a small number of datasets available, which provide thermal video and respiratory rate, but this information was insufficient to fine tune the proposed approach since the loss function requires the estimated and actual signals. To address this gap, we plan to collect a multimodal video dataset for public release in the future.

## 4. Conclusions

This paper presents an end-to-end DL method to estimate respiratory signals and find RR from thermal video data. This study examined 88 recordings from 22 subjects in different conditions: sitting vs. standing and wearing vs. not wearing a mask. In the first stage of the proposed model, a DeTr object detection model is trained to detect the facial ROI. A shallow 3D CNN stage extracts short-term spatio-temporal features from the cropped video. The resulting data are processed by a bi-directional LSTM layer to capture long-term sequential patterns in the data. Our novel model of 3-D CNN and bi-directional LSTM helped the algorithm to capture short-term spatio-temporal and long-term temporal dependencies successively, while keeping the computational complexity small, which is useful for real-time processing. In the last DL step, a fully connected layer connects the bi-directional LSTM results to the output. We also introduced a novel loss function, which is based on negative maximum cross-correlation, and peak frequency difference between the predicted and ground truth signal. The predicted RR is calculated using peak frequency in the output signal. We also compared our method with State-of-the-Art, end-to-end, DL-based methods, including PhysNet, DeepPhys, and StressNet. The results showed that our proposed model outperformed all the aforementioned methods. The proposed method set a new SOTA RR estimation accuracy, while still permitting a real-time RR estimation (inference time is only 5.1 ms for a 10 s window of thermal video data). By calculating the results for each condition, we concluded that our model performed equally well in both conditions of wearing or not wearing a mask. However, we saw that the results for seated subjects were more reliable than for the standing subjects due to reduced body sway. We expect that our proposed method will be applicable in conditions where subjects exhibit limited movement, such as controlled RR monitoring in clinical care facilities.

If only the respiration rate, and not the full respiration signal, is of interest, then one area for future investigation would be to modify the model architecture to directly estimate the respiration rate. The loss function could be modified to place 100% emphasis on the second term, related to the mean absolute error (MAE) between the estimated and actual RR, while de-emphasizing the first loss term, which deals with maximum negative cross-correlation. In this way, model training will be guided solely by RR estimation accuracy. Additionally, it is possible that the model architecture could be further simplified in this case, since regressing the RR is a simpler task than estimating the full respiration signal.

## Figures and Tables

**Figure 1 sensors-24-06386-f001:**
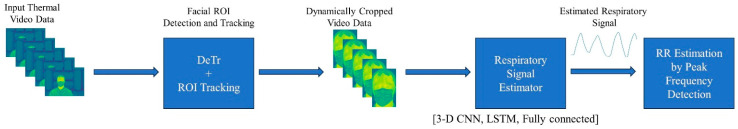
Flowchart of the proposed method. DeTr face detection model finds the face area in each frame. The dynamically cropped frames create a video, which is input into the 3D CNN-based model. RR is calculated by finding the peak frequency of predicted signal.

**Figure 2 sensors-24-06386-f002:**
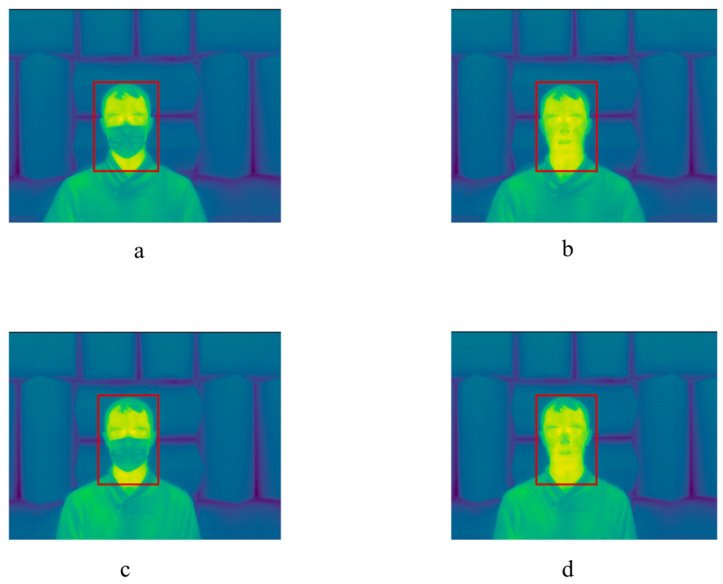
Examples of recorded thermal frames in four conditions: (**a**) sitting with mask, (**b**) sitting without mask, (**c**) standing with mask, and (**d**) standing without mask. The red box indicates the true facial ROI.

**Figure 3 sensors-24-06386-f003:**
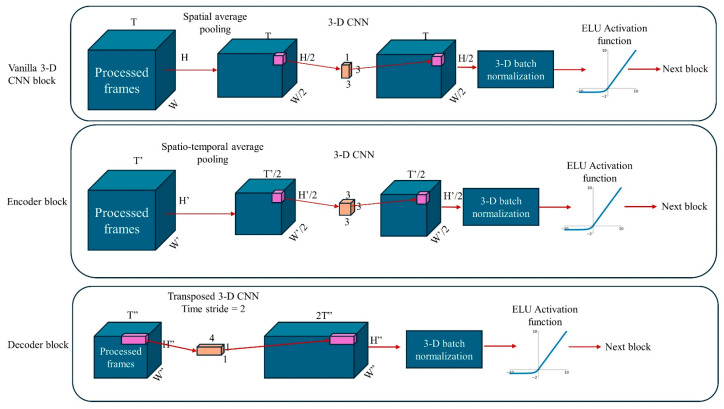
Encoder, decoder, and vanilla 3D CNN block diagrams. Encoder blocks use a temporal stride of 2 during average pooling. Decoder blocks use transposed 3D CNNs with a temporal stride of 2 to restore the sequence length. Vanilla 3D blocks comprise a 3D CNN, with average pooling that does not affect sequence length.

**Figure 4 sensors-24-06386-f004:**
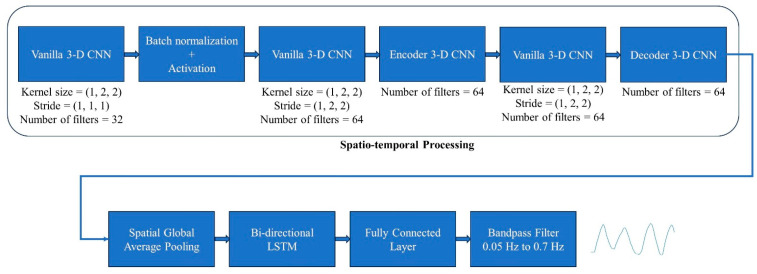
The proposed respiration signal estimation model architecture. The dynamically cropped video goes through encoder, decoder, and vanilla 3D CNN blocks to create 64 feature tensors, each with three dimensions. Spatial average pooling reduces the 3D feature data to 1D (for each feature), and the resulting sequence is processed by a bi-directional LSTM. Finally, a fully connected layer constructs the respiration signal from the output of the bi-directional LSTM stage, which is then bandpass filtered in the range [0.05, 0.7] Hz.

**Figure 5 sensors-24-06386-f005:**
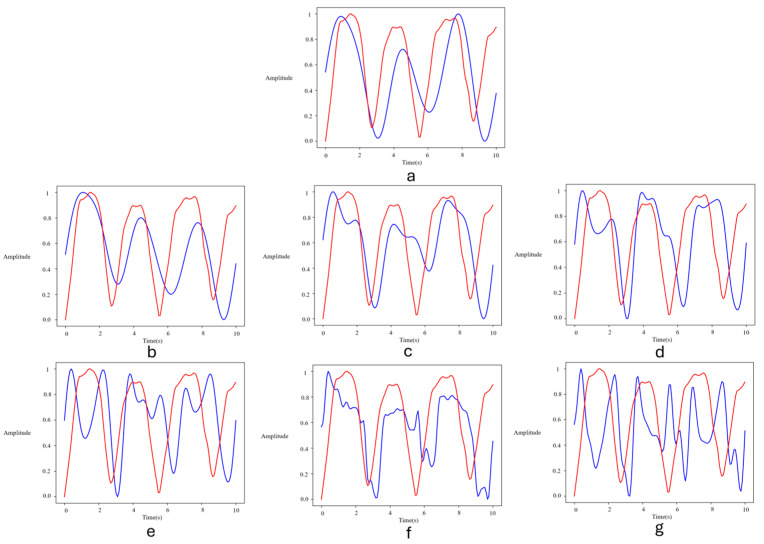
Ground truth respiratory signal (red) and extracted signals (blue) from thermal video data using different models: (**a**) 3D CNN + bi-LSTM; (**b**) 3D CNN + LSTM; (**c**) iBVPNet; (**d**) PhysNet; (**e**) StressNet; (**f**) DeepPhys; and (**g**) averaging. Methods (**a**,**b**) are proposed here. Amplitude units are arbitrary since input and output signals are normalized.

**Figure 6 sensors-24-06386-f006:**
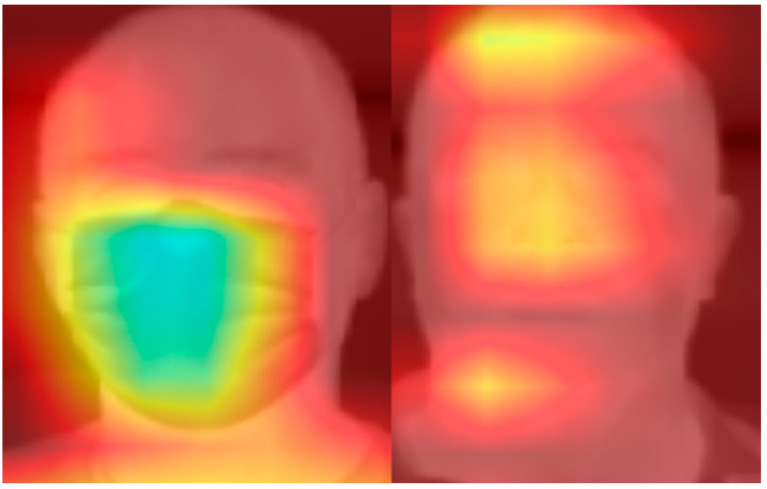
Important areas detected using Grad-CAM method. The left image shows the model’s focus on mask area for estimating respiratory signal. The right figure shows that the nose, head, and neck area are focused on by the model to extract respiratory signal (light areas show the important regions).

**Table 1 sensors-24-06386-t001:** Comparison of the proposed method with existing models for video-based physiological signal estimation. Metrics include RR estimation error, in BPM ± standard deviation; maximum cross correlation between the estimated and actual respiration signal; percentage of clips with <2 BPM error; inference time in ms; and number of learnable parameters inherent to each method.

Method	Error (BPM)	Max *x-corr*	% of Clips with <2 BPM Error	Inference Time (ms)	Learnable Parameters
Spatial Averaging [15]	3.5 ± 4.1	0.57 ± 0.24	29%	0.5	-
DeepPhys * [20]	2.4 ± 0.6	0.71 ± 0.15	54%	3.9	0.8 M
StressNet * [23]	2.4 ± 0.7	0.69 ± 0.15	49%	16	23 M
PhysNet * [22]	2.0 ± 0.4	0.78 ± 0.11	61%	4.2	0.9 M
iBVPNet [22]	1.8 ± 0.5	0.8 ± 0.13	68%	10	1.4 M
3D CNN + LSTM (Developed Here)	1.8 ± 0.4	0.79 ± 0.1	68%	4.5	1 M
3D CNN + bi-LSTM (proposed method)	1.6 ± 0.4	0.81 ± 0.09	71%	5.1	1.3 M

* indicates that these methods were adapted to thermal video-based respiration rate estimation; see text.

**Table 2 sensors-24-06386-t002:** Comparing the results of 3D CNN + bi-LSTM with our proposed and RMSE loss functions. Performance metrics are as in Table 1.

Loss Function	Max *x-corr*	Error in BPM
NMCC + AFD	0.81 ± 0.09	1.6 ± 0.4
RMSE	0.54 ± 0.21	7.7 ± 2.5

**Table 3 sensors-24-06386-t003:** Comparison of model performance in different conditions.

Condition	Error in BPM
Sitting + mask	1.1 ± 0.2
Sitting + no mask	1.5 ± 0.2
Standing + mask	1.8 ± 0.3
Standing + no mask	2.1 ± 0.4

## Data Availability

Research participant video data cannot be shared due to Research Ethics Board constraints. Interested researchers can contact the authors for further information.

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
