# Peer review of "Respiratory Rate Estimation from Thermal Video Data Using Spatio-Temporal Deep Learning"

_sensors, 2024, doi:10.3390/s24196386_

Round 1

Reviewer 1 Report

Comments and Suggestions for Authors

The manuscript has proposed an end-to-end spatio-temporal network to estimate respiration rate from thermal videos. In general, the innovation of the work is not much in my view. I could say that it falls in fine-tuning a deep learning method on new computer vision problem. The core deep learning model remains unchanged, and no significant modifications have been made to the model structure to specifically address the challenge of estimating respiration rate from thermal videos. Therefore, revising the research work reflected in the manuscript will certainly enhance the work to the level of sufficiency in innovation. In its current form the manuscript has some limitations and needs substantial improvement in the following ways:

1- The title of the work does not support its contributions of it. It may be more specific than the proposed one.

2- You should include a broader review of related works on video-based respiration rate estimation, covering both thermal and RGB video approaches, as well as methods that utilize conventional algorithms and deep learning techniques.

3- I am not satisfied with how you elaborate on the contributions of the current work. How do you come up with the proposed approach to estimate respiration rate more accurately than the other methods?

4- Simply reporting the results of the proposed method is insufficient. You should elaborate on the why part of that, with a focus on model interpretability using explainable AI techniques.

5- The proposed evaluation, which exploits data from your private database, seems to indicate a good generalization capacity, on average. In my opinion, it would be interesting to look more closely at whether the method does not work well on some participants, and in this case to try to understand why.

6- No discussion on similar and alternative datasets (if any) present for the task. An evaluation of methods proposed across datasets would emphasize and precipitate their effectiveness and significance.

 7- I recommend to some ablation studies, and reporting results using other metrics such as MAE, RMSE and Pearson’s correlation coefficient. Also, being very specific about whether the experiments deals with general setting or personalized or group-specific settings.

8- I have not noticed any discussion about the limitations of the proposed approach "When will their algorithm fail, etc.?". You should thoroughly provide the weak points of the proposed work in a separate subsection or in Results and Discussion section.

Yours faithfully,

Author Response

Reviewer1

The manuscript has proposed an end-to-end spatio-temporal network to estimate respiration rate from thermal videos. In general, the innovation of the work is not much in my view. I could say that it falls in fine-tuning a deep learning method on new computer vision problem. The core deep learning model remains unchanged, and no significant modifications have been made to the model structure to specifically address the challenge of estimating respiration rate from thermal videos. Therefore, revising the research work reflected in the manuscript will certainly enhance the work to the level of sufficiency in innovation. In its current form the manuscript has some limitations and needs substantial improvement in the following ways:

1- The title of the work does not support its contributions of it. It may be more specific than the proposed one.

We thank the reviewer for suggesting that a more specific title may be appropriate. We have revised the title to specify that deep learning (3D-CNN and LSTM models) are used to achieve spatio-temporal processing of the thermal video data to estimate the respiration signal and rate.

Revised title: Respiratory Rate Estimation from Thermal Video using Spatio-temporal Deep Learning 

2- You should include a broader review of related works on video-based respiration rate estimation, covering both thermal and RGB video approaches, as well as methods that utilize conventional algorithms and deep learning techniques. 

In response to the reviewer’s suggestion to further broaden our literature review, we have added six additional studies related to vital sign estimation from video. Two of these methods leverage deep learning, while the other four leverage traditional digital signal processing approaches. Please see the new discussion of references [11, 12, 13, 14, 22] and [25] in the revised manuscript.

3- I am not satisfied with how you elaborate on the contributions of the current work. How do you come up with the proposed approach to estimate respiration rate more accurately than the other methods?

The justification for our design choices has been further clarified in the revised manuscript (please see Section 1). In essence, our proposed approach differs from traditional approaches (which are largely focused on HR estimation from RGB video) in that we leverage a shallow 3D-CNN stage (to avoid splitting spatial and temporal processing, as done with 2D-CNN + LSTM approaches), while adding a separate LSTM layer to capture longer-term dependencies in the data. Further, we introduce a novel loss function that accounts for the phase shift expected to exist between the ground truth respiration signal measured at the chest, compared to video-based respiration activity from a facial ROI. Lastly, to overcome possible noise in the video and/or ground truth respiration signals, our final loss function combines two terms: the negative maximum cross-correlation between the estimated and actual respiration signal (accounts for lag), and the absolute difference in estimated respiration rate (accounts for noise). This motivation is now clarified on lines 167-175 of the revised manuscript, immediately above our succinct statement of the research contributions made in the present paper (lines 177-186). Our loss function is also more fully motivated on lines 309-312 of the revised manuscript.

4- Simply reporting the results of the proposed method is insufficient. You should elaborate on the why part of that, with a focus on model interpretability using explainable AI techniques.

We thank the reviewer for the suggestion the use of explainable AI techniques to better understand how the models are using the thermal video data to estimate the respiration signal. We have used Grad-CAM [1] to generate visualizations of the spatial regions within the thermal video that contribute most strongly to the respiration signal estimation. Grad-CAM results in the new Figure 6 illustrate that ROIs tend to be in the mask area for masked individuals, and in the nasal area for unmasked individuals. Secondary ROI at the top of the head and the shoulders are likely related to respiration-induced body movement. These results indicate that both variations in temperature, and also respiration-related body movement, contribute to our ability to estimate respiration. In particular, when research participants wore face masks, time-varying thermal patterns dominated the prediction. Please see the new Discussion Section “3.1 Explaining model predictions using Grad-CAM”, and the new Figure 6, in the revised manuscript.

[1]           R. R. Selvaraju, A. Das, R. Vedantam, M. Cogswell, D. Parikh, and D. Batra, “Grad-CAM: Why did you say that?,” Jan. 25, 2017, arXiv: arXiv:1611.07450.

5- The proposed evaluation, which exploits data from your private database, seems to indicate a good generalization capacity, on average. In my opinion, it would be interesting to look more closely at whether the method does not work well on some participants, and in this case to try to understand why.

We appreciate the reviewer’s suggestion to analyze the performance of the proposed method on specific research participants and in specific conditions. This is now discussed in detail in “Section 3.2 Performance on individual subjects and conditions” in the revised manuscript. 

6- No discussion on similar and alternative datasets (if any) present for the task. An evaluation of methods proposed across datasets would emphasize and precipitate their effectiveness and significance.

The examiner makes a good suggestion that validating the methods on an external dataset would further strengthen the paper. Unfortunately, we have been unable to identify a suitable dataset that includes both thermal video of the face and also the ground truth respiratory signal. There are a small number of datasets available that provide thermal video and respiratory rate, but this information is insufficient to fine-tune the proposed approach. We have added a discussion of this point to the new Limitations section of the manuscript and indicated that we plan to collect additional data for public release in the future.

 7- I recommend to some ablation studies, and reporting results using other metrics such as MAE, RMSE and Pearson’s correlation coefficient. Also, being very specific about whether the experiments deals with general setting or personalized or group-specific settings. 

Following the reviewer’s suggestion, we have added a new performance metric reflecting the percent of samples (video clips) that result in RR estimation error  < 2 BPM. Unfortunately, mean absolute error (MAE) and RMSE cannot be used for comparison due to the phase shift between the ground-truth signal, which is measured from the abdomen, and the estimated respiration signal, which is derived from temperature changes in the face.

The results in Table II represent an ablation study, where our proposed method was retrained using a standard loss function, rather than the novel loss function proposed here. Results provide evidence of the efficacy of the proposed loss function.

8- I have not noticed any discussion about the limitations of the proposed approach "When will their algorithm fail, etc.?". You should thoroughly provide the weak points of the proposed work in a separate subsection or in Results and Discussion section.

According to the reviewer’s suggestion, we have added a new section “3.3 Limitations” that specifically addresses the limitations of our study. This discussion supplements the new section (also suggested by this reviewer) on “Section 3.2 Performance on individual subjects and conditions”.

Reviewer 2 Report

Comments and Suggestions for Authors

This paper proposes a deep-learning method for respiration rate prediction from thermal videos. The framework is novel and interesting, but the following comments are suggested to improve the paper’s quality further.

1. More dataset details should be added, such as its size and pre-processing.

2. How did you determine the parameters in the loss function? Why did you set them as 0.5?

3. The evolution criteria should be explained before directly presenting them in tables. 

4. The units of figures should be added.

5. How did you optimize the hyperparameters of your proposed network? The reference Probabilistic framework with Bayesian optimization for predicting typhoon-induced dynamic responses of a long-span bridge should be added to explain different optimization methods.

6. It is suggested to compare more advanced models to illustrate the effectiveness of your proposed method.

Comments on the Quality of English Language

Minor editing of English language required.

Author Response

Reviewer2

  1. More dataset details should be added, such as its size and pre-processing.

We thank the reviewer for the suggestion to provide more information regarding the dataset and its collection. We have carefully revisited the description on lines 193-217. Additionally, we have added two more images to Figure 2 that illustrate all four conditions under which data were collected.

  1. How did you determine the parameters in the loss function? Why did you set them as 0.5?

We have now provided greater guidance to readers on how to set the alpha weight within the proposed loss function. Please see lines 315-322 of the revised manuscript. We did not optimize this parameter for our dataset, and have used a default value of 0.5 which places equal weight on the two loss terms. 

  1. The evolution criteria should be explained before directly presenting them in tables. 

The evaluation metrics are now more clearly defined in the revised caption of Table 1 and in lines 365-383 of the revised manuscript.

  1. The units of figures should be added.

Units are now specified and explained for all figures.

  1. How did you optimize the hyperparameters of your proposed network? The reference Probabilistic framework with Bayesian optimization for predicting typhoon-induced dynamic responses of a long-span bridge should be added to explain different optimization methods.

Thank you for pointing out that we did not discuss hyperparameter optimization. Following your suggestion, we have cited a recent review paper of hyperparameter optimization methods, including Bayesian optimization: [2]. We have additionally included the suggested citation [3] as a specific implementation of Bayesian optimization in an engineering study. Please see lines 320-322 of the revised manuscript.

[2]           B. Bischl et al., “Hyperparameter optimization: Foundations, algorithms, best practices, and open challenges”.

[3]                 Y.-M. Zhang, H. Wang, J.-X. Mao, Z.-D. Xu, and Y.-F. Zhang, “Probabilistic Framework with Bayesian Optimization for Predicting Typhoon-Induced Dynamic Responses of a Long-Span Bridge,” J. Struct. Eng., vol. 147, no. 1, p. 04020297, Jan. 2021.

  1. It is suggested to compare more advanced models to illustrate the effectiveness of your proposed method.

We have already gone to great lengths to include the latest thermal-video-based respiration rate estimation techniques in our comparison with the state of the art. Finding very few such methods in the literature, we have actually included several methods originally intended for estimation of other vital signs (e.g., HR) from other video modalities (e.g., RGB). Those methods required modification to work on the target task of estimating respiration rate from thermal video. However, following the reviewer’s suggestion, we have further expanded our comparison of the state of the art by including iBVPNet [4] (published in 2024) to the revised manuscript. Our method is demonstrated to outperform all five methods included in the comparison with the state of the art.

[4]           J. Joshi and Y. Cho, “iBVP Dataset: RGB-Thermal rPPG Dataset with High Resolution Signal Quality Labels,” Electronics, vol. 13, no. 7, Art. no. 7, Jan. 2024.

Reviewer 3 Report

Comments and Suggestions for Authors

Authors propose an end-to-end deep learning approach to estimate RR from thermal videos. To correct the problem due to the phase shift between the reference and the estimation authors proposed a new loss function. The main contribution of this work is the use of a shallow 3-D CNN stage followed by LSTM stage to estimate the signal. 

The article is well written, the protocol of the experiements are well explained and the comparison to other methods are well described. A plus is to share the codes of the proven methods. 

The only question that the reader can have is because the 3-D CNN + BI-LSTM proposal  does not directly estimate the respiratory frequency and thus avoid the last stage of peaks detection. Please discuss this possibility. 

Author Response

Reviewer 3

Authors propose an end-to-end deep learning approach to estimate RR from thermal videos. To correct the problem due to the phase shift between the reference and the estimation authors proposed a new loss function. The main contribution of this work is the use of a shallow 3-D CNN stage followed by LSTM stage to estimate the signal. 

The article is well written, the protocol of the experiments are well explained and the comparison to other methods are well described. A plus is to share the codes of the proven methods. 

The only question that the reader can have is because the 3-D CNN + BI-LSTM proposal  does not directly estimate the respiratory frequency and thus avoid the last stage of peaks detection. Please discuss this possibility. 

We thank the reviewer for the kind comments and are happy to add a discussion point related to modifying the model architecture to directly estimate the respiration frequency. We have added the following text to the Conclusion section of the revised manuscript:

If only the respiration rate, and not the full respiration signal, is of interest, then one area for future investigation would be to modify the model architecture to directly estimate the respiration rate. The loss function could be modified to place 100% emphasis on the second term related to the mean absolute error (MAE) between the estimated and actual RR, while deemphasizing the first loss term, which deals with maximum negative cross-correlation. In this way, model training will be guided solely by RR estimation accuracy. Additionally, it is possible that the model architecture could be further simplified in this case, since regressing the RR is a simpler task than estimating the full respiration signal.

Round 2

Reviewer 1 Report

Comments and Suggestions for Authors Authors produced a revised version of their manuscript that virtually addresses all the critiques issued to the first submission.